# Zika and dengue but not chikungunya are associated with Guillain-Barré syndrome in Mexico: A case-control study

**Israel Grijalva**[1]*, **Concepción Grajales-Muñiz**[2], **César González-Bonilla**[3], **Victor Hugo Borja-Aburto**[4], **Martín Paredes-Cruz**[1], **José Guerrero-Cantera**[1], **Joaquín González-Ibarra**[3], **Alfonso Vallejos-Parás**[2], **Teresita Rojas-Mendoza**[2], **Clara Esperanza Santacruz-Tinoco**[5], **Porfirio Hernández-Bautista**[2], **Lumumba Arriaga-Nieto**[2], **Ma Guadalupe Garza-Sagástegui**[6], **Ignacio Vargas-Ramos**[7], **Ana Sepúlveda-Núñez**[7], **Omar Israel Campos-Villarreal**[7], **Roberto Corrales-Pérez**[8], **Mallela Azuara-Castillo**[9], **Elsa Sierra-González**[9], **José Alfonso Meza-Medina**[9], **Bernardo Martínez-Miguel**[5], **Gabriela López-Becerril**[1], **Jessica Ramos-Orozco**[10], **Tomás Muñoz-Guerrero**[11], **María Soledad Gutiérrez-Lozano**[11], **Arlette Areli Cervantes-Ocampo**[12]

1 Medical Research Unit for Neurological Diseases, UMAE Hospital de Especialidades, Centro Médico Nacional Siglo XXI, Instituto Mexicano del Seguro Social, Mexico City, Mexico, 2 Division of Epidemiological Surveillance of Communicable Diseases, Epidemiologic Surveillance Coordination, Instituto Mexicano del Seguro Social, Mexico City, Mexico, 3 Division of Surveillance Laboratories and Epidemiological Research, Epidemiologic Surveillance Coordination, Instituto Mexicano del Seguro Social, Mexico City, Mexico, 4 Primary Health-Care Unit. Instituto Mexicano del Seguro Social, Mexico City, Mexico, 5 Central Laboratory of Epidemiology, Centro Médico Nacional La Raza, Instituto Mexicano del Seguro Social, Mexico City, Mexico, 6 Medical Provisions for the State Nuevo León, Instituto Mexicano del Seguro Social, Monterrey, Nuevo León, Mexico, 7 UMAE Hospital de Especialidades No. 25, Instituto Mexicano del Seguro Social, Monterrey, Nuevo León, Mexico, 8 Medical Epidemiological Assistance Coordination of the State of Nuevo León, Instituto Mexicano del Seguro Social, Monterrey, Nuevo León, Mexico, 9 Hospital General de Zona No. 33, Instituto Mexicano del Seguro Social, Monterrey, Nuevo León, Mexico, 10 Hospital General Regional No.1, Instituto Mexicano del Seguro Social, Tijuana, Baja California, Mexico, 11 Hospital General Regional No.1, Instituto Mexicano del Seguro Social Orizaba, Veracruz, Mexico, 12 Hospital General de Zona y Medicina Familiar No. 36, Instituto Mexicano del Seguro Social, Coatzacoalcos, Veracruz, Mexico

* igrijalvao@yahoo.com, israelgrijalva1@gmail.com

**Data Availability Statement:** All relevant data are within the manuscript and its Supporting Information files.

## Abstract

### Background

Zika, dengue and chikungunya viruses (ZIKV, CHIKV and DENV) are temporally associated with neurological diseases, such as Guillain-Barré syndrome (GBS). Because these three arboviruses coexist in Mexico, the frequency and severity of GBS could theoretically increase. This study aims to determine the association between these arboviruses and GBS in a Mexican population and to establish the clinical characteristics of the patients, including the severity of the infection.

A case-control study was conducted (2016/07/01-2018/06/30) in Instituto Mexicano del Seguro Social (Mexican Social Security Institute) hospitals, using serum and urine samples that were collected to determine exposure to ZIKV, DENV, CHIKV by RT-qPCR and serology (IgM). For the categorical variables analysis, Pearson's $\chi^2$ or Fisher exact tests were used, and the Mann-Whitney U test for continuous variables. To determine the association of GBS and viral infection diagnosis through laboratory and symptomatology before admission, we calculated the odds ratio (OR) and 95% confidence intervals (95%CI) using a 2x2

**Funding:** This project was totally funded by the Instituto Mexicano del Seguro Social (https://www. imss.gob.mx) through Fondo de Investigación en Salud, grant number FIS/IMSS/PROT/PRIO/16/ 055. The grant was obtained by IG. The funders had no role in study design, data collection and analysis, decision to publish, or preparation of the manuscript.

**Competing interests:** The authors have declared that no competing interest exist.

contingency table. A $p$-value $\leq 0.05$ was considered as significant. Ninety-seven GBS cases and 184 controls were included. The association of GBS with ZIKV acute infection (OR, 8.04; 95% CI, 0.89–73.01, p = 0.047), as well as laboratory evidence of ZIKV infection (OR, 16.45; 95% CI, 2.03–133.56; p = 0.001) or Flavivirus (ZIKV and DENV) infection (OR, 6.35; 95% CI, 1.99–20.28; p = 0.001) was observed. Cases of GBS associated with ZIKV demonstrated a greater impairment of functional status and a higher percentage of mechanical ventilation. According to laboratory results, an association between ZIKV or ZIKV and DENV infection in patients with GBS was found. Cases of GBS associated with ZIKV exhibited a more severe clinical picture. Cases with co-infection were not found.

## Author summary

Dengue (DENV), chikungunya (CHIKV), and Zika (ZIKV) are considered as emerging or re-emerging viruses. Recently, these viruses have produced major epidemics in tropical climate urban centers, and have been associated with neurological manifestations, including Guillain-Barre syndrome (GBS), which causes muscle weakness, unstable gait, and decreased or absent musculoskeletal reflexes. This study aims to investigate the association between these viral infections and GBS. A case and control study was conducted nationwide, including 97 cases of GBS and 184 controls matched by age, gender, and locality, but not the disease. The study shows a positive association between GBS cases and ZIKV or ZIKV and DENV infection. GBS cases associated with ZIKV depicted a more severe clinical picture (more impairment of functional status, incapacity, and a higher percentage of mechanical ventilation). Finally, the symptoms of suspected ZIKV disease prior to the development of GBS were similar to some previous reports. The impact of the interaction of these three arboviruses, particularly ZIKV, on the health of the Mexican population was less than expected. The Mexican experience could be useful for other populations.

## Introduction

Arboviruses are a significant cause of human disease worldwide [1]. Viruses such as Zika (ZIKV), dengue (DENV) and chikungunya (CHIKV) lead to major epidemics in tropical climate urban centres [1] and have expanded recently, resulting in outbreaks of considerable scale in the Western Hemisphere [2].

Several neurological manifestations, including Guillain-Barré syndrome (GBS), are associated with ZIKV, DENV or CHIKV [3–5]. Moreover, these arboviruses have been found to produce co-infection related to neurological diseases in some patients [6,7].

GBS is an acute immune-mediated inflammatory polyradiculoneuropathy that typically produces predominantly distal symmetrical muscular weakness, unsteady gait, and hyporeflexia or areflexia [8]. About two-thirds of patients with GBS have reported a history of *Campylobacter jejuni*, cytomegalovirus, Epstein-Barr virus or *Mycoplasma pneumoniae*, as well as influenza A virus and *Haemophilus influenzae* infections. Recently, an association with ZIKV, CHIKV and DENV has been reported as well [3–5,8,9]. Mortality remains around 2.6% in the USA [10], yet a mortality rate of 5–10% was documented in 2014 in Mexico [11].

We know that these three arboviruses have coexisted in Mexico since 2015. On the one hand, the pre-existing immunity to DENV could strengthen ZIKV infection (both flaviviruses) by increasing the severity of the disease or co-infections [6,7,12,13]. On the other hand, due to their coexistence and interactions, a probability of predominance of any of the three

arboviruses exists—since the Mexican population had not been exposed to ZIKV earlier. Considering these facts, the aim of this study was to identify the association of GBS with these arboviruses in a Mexican population, the demographic and clinical characteristics of the patients, the severity of the infection and the presence of co-infection.

## Methods

### Ethics statement

The study protocol was approved by the National Committee for Scientific Research (R-2016-785-026), under the Instituto Mexicano del Seguro Social (IMSS). Individuals were invited to participate in the study and were given informed consent forms. After being provided with the necessary information and their questions clarified, all the participants who met the inclusion criteria (cases and controls) signed an informed consent form to participate in the study. A parent or guardian of any minor participating provided written informed consent on their behalf.

### Study design and participants

A case-control study was conducted to identify the role of ZIKV, DENV and CHIKV infection in the development of GBS in Mexico, from July 1, 2016, to June 30, 2018. GBS patients and control subjects were selected from IMSS hospital units nationwide.

### Cases

All cases were patients developing GBS who were identified prospectively by a second level physician due to flaccid paralysis. If necessary, patients were sent to a third–level–care hospital. Patients were evaluated by a neurologist, who diagnosed GBS based on the clinical features and neurological manifestations of the disease, as well as cerebrospinal fluid (CSF) analysis and electrodiagnostic test results, in keeping with international criteria [14]. Medical records were reviewed and diagnostic tests—including CSF, neuroimaging, and electrophysiological studies when available—were conducted to establish the clinical characteristics of the disease and confirm the GBS diagnosis. Patients who met levels 1–3 of diagnostic certainty according to the Brighton Collaboration criteria were considered as eligible and were invited to participate in the protocol.

### Controls

Controls were identified by nursing personnel hired to perform this task from the lists of inpatients with a non-febrile disease (no fever documented 48 hours before enrolment) or patients for elective surgery in the same hospital, or healthy relatives of these cases, who met the selection criteria. Controls were selected as a non-probability convenience sampling of individuals paired by age (± 5 years), gender, and place of residence with the corresponding case, and enrolled within seven days from the date of inclusion of the GBS case. Two controls were included for each GBS-case recruited.

Clinical and demographic data were collected directly from the patients or their relatives. Patient information included age, gender, place of residence, clinical history of comorbidities, signs and symptoms, duration and severity of the disease based on the GBS Disability Score and Medical Research Council Scale for Muscle Strength (MRC) and treatment. The neurophysiological evaluation included motor and sensory nerve conduction studies of upper and lower extremity peripheral nerves. A lumbar puncture was performed by standard technique to obtain CSF for cytological and cytochemical analysis.

All the participants were interviewed by nursing personnel hired and trained to perform this task to seek more information about risk factors and exposure during the last two months, before the onset of the neurological symptoms for cases or in general the last two months for controls. Consequently, individuals were invited to participate in the study and were given informed consent forms. Once they agreed to participate and signed the consent, they were included in the study to continue with the corresponding procedures.

After the interviews, serum and urine samples were collected to determine exposure to ZIKV, DENV and CHIKV during the first seven days after the onset of the neurological picture. If the individuals agreed to participate in the study, blood and urine samples were collected from each control as well.

The following definitions were established to determine the type of ZIKV, DENV or CHIKV infection:

Acute infection by ZIKV, DENV, or CHIKV was defined as a positive result to the reverse transcriptase real-time polymerase chain reaction (RT-qPCR) in either serum or urine samples for ZIKV, or positive serum samples for DENV or CHIKV as follows:

- Positive ZIKV and negative DENV and CHIKV results: acute ZIKV infection.

- Positive DENV and negative ZIKV and CHIKV results: acute DENV infection.

- Positive CHIKV and negative ZIKV and DENV results: acute CHIKV infection.

- Positive ZIKV, DENV, or CHIKV results: co-infection.

Laboratory evidence of ZIKV, DENV or CHIKV infection was defined as a positive ZIKV, DENV, or CHIKV RT-qPCR result in serum or urine samples, or an anti-ZIKV, anti-DENV or anti-CHIKV immunoglobulin M (IgM) detected in serum as follows:

- Positive ZIKV RT-qPCR or positive ZIKV IgM and negative DENV RT-qPCR or negative DENV IgM and negative CHIKV RT-qPCR or negative CHIKV IgM results: laboratory evidence of ZIKV infection.

- Positive DENV RT-qPCR or positive DENV IgM and negative ZIKV RT-qPCR or negative ZIKV IgM and negative CHIKV RT-qPCR or negative CHIKV IgM results: laboratory evidence of DENV infection.

- Positive CHIKV RT-qPCR or positive CHIKV IgM and negative ZIKV RT-qPCR or negative ZIKV IgM and negative DENV RT-qPCR or negative DENV IgM results: laboratory evidence of CHIKV infection.

- Positive ZIKV RT-qPCR or positive ZIKV IgM and positive DENV RT-qPCR or positive DENV IgM and negative CHIKV RT-qPCR or negative CHIKV IgM results: laboratory evidence of flavivirus infection.

- Positive ZIKV RT-qPCR or positive ZIKV IgM and positive DENV RT-qPCR or positive DENV IgM and positive CHIKV RT-qPCR or positive CHIKV IgM results: laboratory evidence of arbovirus infection.

## Laboratory analysis

Diagnostic tests were performed at the Central Laboratory of Epidemiology, La Raza National Medical Center, IMSS, Mexico City, following the epidemiological surveillance guidelines and in compliance with the national regulations established by the Mexican Ministry of Health. Tests were performed sequentially: first, RT-qPCR for each virus. If the RT-

qPCR result was negative, IgM antibody determination against ZIKV, DENV and CHIKV was performed.

Viral RNA extraction from serum and urine samples was carried out using the QIAmp Viral RNA Mini Kit (Cat. 52906, Qiagen, Hilden, Germany) or the MagNA Pure LC Total Nucleic Acid Isolation Kit (Cat. 03038505001, Roche Diagnostic, Basel, Switzerland) following the manufacturer's recommendations.

## RT-qPCR

The detection of RNA from ZIKV, CHIKV and DENV was carried out by RT-qPCR, using the uniplex or triplex format.

In the case of ZIKV, primers and probes were used to amplify the coding region of protein E as previously reported [15]. Amplification of CHIKV was performed with primers and probes, including positions 6856 to 6891 of the structural region of the genome [16], while for DENV, primers and probes were designed to amplify the NS5 non-structural protein [17].

The uniplex RT-qPCR assays to detect DENV were performed with the QuantiTect Probe RT-PCR Kit (Cat. 204445, Qiagen, Hilden, Germany), and the amplification of RT-PCR was performed with Superscript III Platinum One-Step reagents RT-qPCR System (Cat. 11732–088, Invitrogen, Carlsbad, CA, USA). The Applied biosystems 7500 fast platform was used in all cases. SDS software version 1.4 was used for the analysis of the results.

For the detection of RNA from ZIKV, CHIKV and DENV by trioplex RT-qPCR, the commercial VIASURE Zika, Dengue & Chikungunya Real-Time PCR Detection Kit (Cat. VS-ZDC112L, Viasure, Zaragoza, Spain) was used and implemented in the Applied biosystems 7500 fast platform and the SDS software version 2.3.

## Serology

Serum samples were processed for the determination of IgM antibodies against ZIKV and DENV by commercial qualitative methods using the commercial kits Anti-Zika Virus ELISA IgM (Cat. EI2668-9601M, Euroimmun AG, Lübeck, Germany), as well as Panbio Dengue IgM Capture ELISA (Cat. 01PE20/01PE21, Panbio Diagnostics, Korea) and following the indications of the manufacturer. The results were evaluated together to discard cross-reactivity between flaviviruses (ZIKV and DENV).

The detection of IgM antibodies against Chikungunya was carried out with the reagent CHIKjj Detect IgM ELISA (Cat. CHKM-C, InBios International, Inc. USA), following the instructions of the manufacturer.

## Statistical analysis

The sample size was calculated considering published data by Cao-Lormeau et al. [9] Descriptive statistics were used for demographic and clinical data. Categorical variables were presented as counts and proportions and continuous variables as mean and standard deviation. For categorical variables, Pearson's $\chi^2$, or Fisher exact tests were used, and the Mann-Whitney U test was used for continuous variables. For the association of GBS with viral infection diagnosis by laboratory and with symptomatology before admission, odds ratio (OR) and 95% confidence intervals (95%CI) were calculated using a 2x2 contingency table. In all cases, a $p$-value $\leq 0.05$ was considered as statistically significant. IBM SPSS v22 software was used for the statistical analysis.

## Results

For two years, ninety-seven cases and 184 control subjects fulfilled selection criteria and were recruited (87 cases with two controls and 10 cases with one control). Patients with GBS and control subjects from nine States of Mexico were included.

The mean age of patients with GBS and controls was 39 ± 19.3 years and 39.7 ± 18.5 years, respectively. Males were predominant in both groups (59/87 and 113/184, respectively), representing 61% of the total participants.

### Laboratory determinations

Taking information from all laboratory analyses (RT-qPCR in serum and urine and IgM in serum) eight ZIKV, four DENV and one CHIKV positive cases were found. ZIKV+ cases came from Veracruz and Nuevo León, and DENV+ cases came from Veracruz and Baja California. The CHIKV+ case came from Nuevo León. As for the controls, one ZIKV+ came from Veracruz), three DENV+ from Nuevo León and six CHIK+ from Veracruz (Table 1).

From eight ZIKV+ cases, four were identified by RT-qPCR (two from serum and two from urine samples) and four were identified by IgM, of which six were DENV- and CHIKV-. Unfortunately, the remaining two samples could not be analysed due to insufficient volume. All four DENV+ cases were determined by IgM; neither was positive for ZIKV nor CHIKV. The only CHIK+ case identified by IgM was negative for both ZIKV and DENV.

Using RT-qPCR, a positive ZIKV acute infection was demonstrated in four cases (two in serum and two in urine) and one control (urine) (OR, 8.04; 95% CI, 0.89–73.01; p = 0.047). In contrast, no positive DENV acute infections were found in either cases or controls. Interestingly, one positive CHIKV from the control group was identified, whereas none from the cases (Table 2).

Laboratory evidence of ZIKV infection was observed in eight GBS cases (four by RT-qPCR and four by IgM) and one control (by RT-qPCR) (OR, 16.45; 95% CI, 2.03–133.56; p = 0.001). No significant differences were observed when considering DENV+ or CHIKV+ results. However, when flaviviruses associated with GBS (ZIKV+ and DENV+) were considered together (12 cases = 8 ZIKV+ and 4 DENV+, and 4 controls = 1 ZIKV+ and 3 DENV+), a significant difference was observed (OR, 6.35; 95% CI, 1.99–20.28; p = 0.001). Laboratory evidence of arbovirus infection (OR, 2.69; 95% CI, 1.13–6.39; p = 0.037) was also found.

### Neurologic alterations

All cases (97) were classified according to Brighton criteria (16 in level 1; 39 in level 2; and 42 in level 3). The neurological symptomatology at the nadir of all GBS cases, ZIKV+ and ZIKV- is shown in Table 3.

**Table 1. Distribution of positive laboratory results for ZIKV, DENV and CHIKV (RT-qPCR or IgM) in GBS cases (n = 97) and control subjects (n = 184) in 9 states of Mexico (July 2016 –June 2018).**

| State | Cases | ZikaV + | DenV+ | ChikV+ | Controls | ZikaV + | DenV+ | ChikV+ |
|---|---|---|---|---|---|---|---|---|
| Baja California | 10 | 0 | 1 | 0 | 19 | 0 | 0 | 0 |
| Campeche | 1 | 0 | 0 | 0 | 2 | 0 | 0 | 0 |
| Morelos | 1 | 0 | 0 | 0 | 2 | 0 | 0 | 0 |
| Chiapas | 1 | 0 | 0 | 0 | 2 | 0 | 0 | 0 |
| Nayarit | 1 | 0 | 0 | 0 | 2 | 0 | 0 | 0 |
| Nuevo León | 37 | 3 | 0 | 1 | 72 | 0 | 3 | 0 |
| Tabasco | 1 | 0 | 0 | 0 | 2 | 0 | 0 | 0 |
| Quintana Roo | 1 | 0 | 0 | 0 | 1 | 0 | 0 | 0 |
| Veracruz | 44 | 5 | 3 | 0 | 82 | 1 | 0 | 6 |
| Total | 97 | 8 | 4 | 1 | 184 | 1 | 3 | 6 |

**Table 2. Laboratory analyses in GBS cases and controls for ZIKV, DENV or CHIKV infection according to the techniques used (RT-qPCR for serum and urine samples and IgM in serum).** Mexico, July 2016-June 2018.

| Viral infection | | Cases | | | Controls | | | OR (CI 95%) | p |
|---|---|---|---|---|---|---|---|---|---|
| | Technique | Negative | Positive | Total Analysed[1] | Negative | Positive | Total Analysed[1] | | |
| **ACUTE VIRAL INFECTION** | | | | | | | | | |
| ZIKV | RT-qPCR[2] | 91 | 4 | 95 | 183 | 1 | 184 | 8.04 (0.89–73.01) | 0.047 |
| DENV | RT-qPCR[3] | 84 | 0 | 84 | 172 | 0 | 172 | - | - |
| CHIKV | RT-qPCR[3] | 84 | 0 | 84 | 171 | 1 | 172 | - | - |
| **LABORATORY EVIDENCE OF VIRAL INFECTION** | | | | | | | | | |
| ZIKV | RT-qPCR[2] or IgM | 89 | 8 | 97 | 183 | 1 | 184 | 16.45 (2.03–133.56) | 0.001 |
| DENV | RT-qPCR or IgM | 82 | 4 | 86 | 169 | 3 | 172 | 2.75 (0.60–12.56) | 0.226 |
| CHIKV | RT-qPCR or IgM | 85 | 1 | 86 | 166 | 6 | 172 | 0.33 (0.04–2.75) | 0.430 |
| Flavivirus (ZIKV and DENV) | RT-qPCR or IgM | 85 | 12 | 97 | 180 | 4 | 184 | 6.35 (1.99–20.28) | 0.001 |
| Arbovirus (ZIKV and/or DENV and CHIKV) | RT-qPCR or IgM | 84 | 13 | 97 | 174 | 10 | 184 | 2.69 (1.13–6.39) | 0.037 |

[1] Not mutually exclusive

[2] ZIKV RT-qPCR includes serum and urine samples

[3] Undefined odds ratio/confidence limits/significance. Flavivirus co-infection and arbovirus co-infection (ZIKV or DENV and CHIKV) are not presented because of the absence of positive cases for DENV and CHIKV.

Significant differences can be observed between ZIKV+ and ZIKV- cases in oculomotor and global hypotonia ($p = 0.02$ and 0.02, respectively) but not in facial paresis ($p = 0.14$).

Furthermore, a significant impairment was observed in some clinical outcomes between ZIKV+ and ZIKV-: MRC at discharge ($p < 0.01$), GBS Disability Score at discharge ($p = 0.02$), and Modified Rankin Scale mainly at discharge ($p = 0.02$) (Table 4).

**Table 3. Comparison of the main signs and symptoms in the nadir of the clinical evolution of Guillain-Barré syndrome (GBS), ZIKV+, ZIKV- and all GBS cases.** Mexico, July 2016-Jun 2018.

| Signs and symptoms | A All GBS cases n = 97 | | B ZIKV+ n = 8 | | C ZIKV - n = 89 | | p-value |
|---|---|---|---|---|---|---|---|
| | Frequency | (%) | Frequency | (%) | Frequency | (%) | B versus C |
| Oculomotor dysfunction | 19 | 19.5 | 4 | 50 | 15 | 16.9 | 0.02 |
| Facial paresis | 32 | 32.9 | 5 | 62.5 | 27 | 30.3 | 0.14 |
| Bulbar dysfunction | 21 | 21.6 | 4 | 50 | 17 | 19.1 | 0.06 |
| Upper limb weakness | 74 | 76.3 | 6 | 75 | 68 | 76.4 | 0.60 |
| Lower limb weakness | 89 | 91.8 | 8 | 100 | 81 | 91.0 | 0.83 |
| Generalized hypotonia | 45 | 46.4 | 7 | 87.5 | 38 | 42.7 | 0.02 |
| Dizziness | 4 | 4.1 | 1 | 12.5 | 3 | 3.4 | 0.29 |
| Ataxia | 21 | 21.6 | 2 | 25 | 19 | 21.3 | 0.55 |
| Lower back pain | 3 | 3.1 | 1 | 12.5 | 2 | 2.2 | 0.22 |
| Respiratory distress | 20 | 20.6 | 3 | 37.5 | 17 | 19.1 | 0.35 |
| Sensory alterations | 37 | 38.1 | 3 | 37.5 | 34 | 38.2 | 0.64 |
| Palpitations | 2 | 2.1 | 0 | 0 | 2 | 2.2 | 0.84 |
| Low blood pressure | 2 | 2.1 | 0 | 0 | 2 | 2.2 | 0.84 |

**Table 4. Clinical characteristics and outcome of ZIKV+, ZIKV- and all cases during the evolution of the Guillain-Barré syndrome in Mexico.** July 2016-June 2018.

| Clinical characteristics and outcome | A<br>All GBS cases<br>n = 97 | | B<br>ZIKV+<br>n = 8 | | C<br>ZIKV -<br>n = 89 | | p-value |
|---|---|---|---|---|---|---|---|
| | Mean | SD | Mean | SD | Mean | SD | B versus C |
| Latency (days) | 5.64 | 6.77 | 4.13 | 3.48 | 5.79 | 7.04 | 0.51 |
| Hospital stay (days) | 11.97 | 10.34 | 15.43 | 9.41 | 11.56 | 10.45 | 0.35 |
| MRC (admission) | 32.66 | 17.47 | 23.5 | 13.3 | 33.71 | 17.64 | 0.10 |
| MRC (discharge) | 42 | 17.13 | 20.67 | 14.67 | 41.40 | 16.52 | *<0.01* |
| GBS disability score (admission) | 3.85 | 0.91 | 4.25 | 0.70 | 3.81 | 0.93 | 0.20 |
| GBS disability score (discharge) | 3.15 | 1.13 | 4 | 1.41 | 3.05 | 1.06 | *0.02* |
| Modified Rankin Scale (admission) | 3.98 | 0.96 | 4.63 | 0.51 | 3.90 | 0.97 | *0.04* |
| Modified Rankin Scale (discharge) | 3.25 | 1.17 | 4.13 | 1.45 | 3.15 | 1.10 | *0.02* |
| | Frequency | % | Frequency | % | Frequency | % | |
| IV Immunoglobulin | 81 | 83.51 | 6 | 75 | 75 | 84.27 | 0.61 |
| Plasmapheresis | 2 | 2.06 | 0 | 0 | 2 | 2.25 | 0.84 |
| Mechanical ventilation | 17 | 17.53 | 3 | 42.9 | 14 | 15.73 | 0.14 |
| Nosocomial pneumonia | 2 | 2.06 | 1 | 12.5 | 1 | 1.12 | 0.15 |
| ICU management | 16 | 16.49 | 1 | 12.5 | 15 | 16.85 | 0.60 |
| Mortality | 4 | 4.12 | 1 | 12.5 | 3 | 3.37 | 0.29 |

GBS: Guillain Barré Syndrome Disability Sum Score; ICU, intensive care unit; MRC: Medical Research Council Scale for Muscle Strength Sum Score.

Intravenous immunoglobulin (81 cases) and plasmapheresis (two cases) were administered as a treatment. Mechanical ventilation was required in 17 of all the GBS patients (17%), of which three were ZIKV+ (3/8, 42.9%) and fourteen ZIKV- (14/89, 15.73%) with no statistical significance. Other clinical outcomes of GBS patients are shown in Table 4.

Forty electrophysiological studies were conducted, of which four were ZIKV+ cases (three of the axonal type, acute motor-sensory axonal neuropathy [AMSAN] subtype, and one of the axonal type, acute inflammatory demyelinating polyneuropathy [AIDP] subtype); thirty-six were ZIKV- cases: 28 of the axonal type (15 AMSAN and 13 acute motor axonal neuropathy [AMAN] subtypes), and eight of the demyelinating type (AIDP subtype).

Suspected case symptoms of ZIKV before the development of GBS in ZIKV+ cases, such as rash, fever and conjunctival hyperemia, were more frequent and significantly different than controls. Other significant symptoms are shown in Table 5.

## Discussion

The present study results show an association between GBS and ZIKV, as well as between GBS and ZIKV/DENV, both flaviviruses, supported by laboratory analysis. Moreover, higher severity of GBS is associated with ZIKV and co-infections were not demonstrated among these three viruses.

Several reports indicate the probability of a causal association between ZIKV and GBS due to the increase in the number of cases of GBS during ZIKV outbreaks [18–20]. Some describe a greater severity of GBS symptoms [12,13,21]. However, only eight reports with an appropriate design to show a positive or negative association were found [9,21–27].

In the present work, viral RNA showing ZIKV acute infection was significant ($p = 0.047$). Despite significant differences between positive ZIKV cases and controls, the lower endpoint of the confidence interval is $< 1$; consequently, we were unable to demonstrate an association between ZIKV acute infection and GBS. At present, only one report has demonstrated an acute ZIKV infection associated with GBS by RT-PCR, in Puerto Rico [22].

**Table 5. Symptoms prior to the development of GBS in cases (positive or negative to ZIKV) and controls in México.** July 2016-June 2018.

| Symptoms | Controls n = 184 | | GBS cases n = 97 | | OR (CI 95%) | p-value | ZIKV+ n = 8 | | ZIKV- n = 89 | |
|---|---|---|---|---|---|---|---|---|---|---|
| | Frequency | % | Frequency | % | | | Frequency | % | Frequency | % |
| Rash | 2 | 1 | 22 | 22.7 | 26.69 (6.12–116.36) | p<0.01 | 3 | 37.5 | 19 | 21.4 |
| Conjunctival hyperemia | 2 | 1 | 18 | 18.6 | 20.73 (4.69–91.49) | p<0.01 | 3 | 37.5 | 15 | 16.9 |
| ADD | 2 | 1 | 13 | 13.4 | 16.34 (3.59–74.36) | p<0.01 | 0 | 0 | 13 | 14.6 |
| Retro-ocular pain | 2 | 1 | 11 | 11 | 11.64 (2.52–53.66) | p<0.01 | 3 | 37.5 | 8 | 8.9 |
| Myalgias | 12 | 6.5 | 41 | 42 | 10.49 (5.15–21.35) | p<0.01 | 4 | 50 | 37 | 41.6 |
| Diarrhoea | 10 | 5.4 | 35 | 36 | 9.82 (4.59–21.00) | p<0.01 | 2 | 25 | 33 | 37.1 |
| Arthralgias | 12 | 6.5 | 39 | 40 | 9.63 (4.72–19.64) | p<0.01 | 3 | 37.5 | 36 | 40.4 |
| Malaise | 16 | 8.7 | 46 | 47.4 | 9.47 (4.94–18.13) | p<0.01 | 4 | 50 | 42 | 47.2 |
| Pruritus | 3 | 1.6 | 12 | 12 | 8.51 (2.34–30.97) | p<0.01 | 3 | 37.5 | 9 | 10.1 |
| Headache | 21 | 11.4 | 49 | 50.5 | 7.92 (4.31–14.49) | p<0.01 | 4 | 50 | 45 | 50.6 |
| Fever | 18 | 9.8 | 44 | 45 | 7.65 (4.07–14.36) | p<0.01 | 5 | 62.5 | 39 | 43.8 |
| Sore throat | 6 | 3.3 | 14 | 14.4 | 5.00 (1.85–13.48) | p<0.01 | 3 | 37.5 | 11 | 12.4 |
| ARI | 8 | 4.3 | 13 | 13.4 | 3.74 (1.48–9.43) | p<0.01 | 1 | 12.5 | 12 | 13.5 |
| Cough | 11 | 6 | 16 | 16.5 | 3.10 (1.38–6.99) | p<0.01 | 1 | 12.5 | 15 | 16.9 |
| Vomiting | 8 | 4.3 | 11 | 11 | 2.81 (1.09–7.25) | p 0.02 | 0 | 0 | 11 | 12.4 |
| Nausea | 9 | 5 | 12 | 12.4 | 2.74 (1.11–6.76) | p 0.02 | 0 | 0 | 12 | 13.5 |
| Abdominal pain | 17 | 9.2 | 21 | 21.6 | 2.71 (1.35–5.43) | p<0.01 | 1 | 12.5 | 20 | 22.5 |
| Other symptoms with no statistical significance (1) | | | | | | | | | | |

ADD: acute diarrhoeal diseases; ARI: acute respiratory infections; OR: odds ratio; CI 95%: confidence interval 95%; (1) Expectoration, nasal congestion, rhinorrhea constipation, coryza, lymphadenopathy, chest pain, pelvic pain, abdominal distension.

When considering the laboratory evidence of ZIKV infection by RT-PCR (serum or urine samples) or IgM anti-ZIKV (serum), our findings demonstrated an association with GBS, similar to Dirlikov et al., in Puerto Rico [22], and Simon et al., in New Caledonia [25]. Moreover, the previous infection by ZIKV was also considered through the combination of positive results with IgM and IgG anti ZIKV, along with the neutralising response against ZIKV, described by Cao-Lomeau et al. in French Polynesia, demonstrating the association between GBS and ZIKV for the first time [9].

In contrast, two of the eight case-control studies previously mentioned did not demonstrate any association [21,26].

Concerning GBS secondary to acute DENV infection identified by RT-PCR, most publications indicate an apparent association [5,28,29]. Unfortunately, a causal association has not been convincingly demonstrated. In the present research, this association could not be demonstrated either. However, a significant association between GBS and flavivirus (ZIKV and DENV) was demonstrated, considering the laboratory evidence of virus infection. These findings agree with those previously reported by Salinas et al. [23], Styczynski et al. [24] and Simon et al. [25] in other populations (Colombia, Brazil, and New Caledonia, respectively). Although no conclusive role of DENV in the triggering of GBS has been demonstrated, an association between GBS and flavivirus (ZIKV and DENV) seems to be established.

No association between GBS and CHIKV could be demonstrated, probably due to the small sample size.

Finally, no co-infection was demonstrated since only four positive ZIKV RT-qPCR cases were found and no positive cases for DENV or CHIKV, in contrast to other previously described results [6,7].

We have only taken into account the eight published case and control studies for the purpose of discussion for this paper, in spite of the comments published point out their limitations, such as: lack of precision in the clinical and differential diagnoses, flaws in the methodological design, biases and statistical limitations. Despite the foregoing, we consider this type of design to be the appropriate one to determine whether or not we are on the right track to associate GBS with any of the viral infections under study [18,30,31].

## Neurological behaviour

GBS associated with ZIKV is often related to dysautonomia, paralysis of cranial nerves and more rapid onset of signs and symptoms [20,21]. However, in the present work, no dysautonomia was observed, but the paralysis or paresis of oculomotor nerves and generalised hypotonia were the most significant clinical alterations observed. The need for ventilator support varies from 9.1 to 28%, according to other studies [10,14,32]. By contrast, it was more common in the present work (42.9%). Although the difference was not statistically significant, it is consistent with previous reports [21]. A higher percentage of patients with muscular weakness, impairment of functional status and incapacity, as well as a higher percentage of mechanical ventilation shown in the present study, indicate that GBS cases associated with ZIKV may be more severe than those seen with other antecedent aetiologies of GBS [33,34].

When considering the electrophysiological results of the present cases, the AMSAN subtype was more frequently observed, in keeping with what has been reported for the Asia/Asia Pacific (French Polynesia) [9] and the Americas (Cúcuta, Colombia) [20] regions, where the axonal type also predominated, though it was the AMAN subtype. On the contrary, these results differ from other regions of Asia, including Bangladesh and the Americas (Cúcuta and Barranquilla, in Colombia) as well, in which the demyelinating type AIDP subtype was the most frequent [21,23,26]. Interestingly, both AMAN and AIDP cases were found in Northeastern Mexico [27].

In this work, symptoms prior to the development of GBS related to the symptoms of ZIKV suspected disease, such as rash, fever and conjunctivitis, showed significant differences to controls. Other authors have highlighted these symptoms as symptoms of suspicion [23,24]. In 2016, Cao-Lormeau et al. described these symptoms (rash, arthralgia and fever); however, they did not analyse the risk [9]. Some recent studies have also reported this association: Salinas et al. (2017) reported rash, fever, arthralgia and myalgia [23]; Styczynski et al. (2017) reported rash, headache, fever, arthralgias and myalgias [24]. Interestingly, Gongora-Rivera et al. [27] found typical ZIKV symptoms (rash, joint pain and conjunctivitis) associated with laboratory evidence of ZIKV infection. When considering the symptoms reported in these studies as well as our own findings, the most significant symptoms are rash, arthralgia, fever, myalgia, conjunctivitis and headache.

Considering that ZIKV, DENV, and CHIKV produce large epidemics worldwide [1], and these three arboviruses coexist in Mexico since 2015 [35], this study confirms their presence in both GBS cases and controls. However, only ZIKV or DENV but no CHIKV were associated with GBS. Similar to previous studies, the present report also demonstrated that GBS cases associated with ZIKV showed a more severe clinical picture [12,13], which should be considered for the treatment of these patients in countries where previous ZIKV infection could lead to this neurological disease. Although co-infections between these arboviruses have been described previously [6,7], cases in which two or three of these viruses were simultaneously present were not demonstrated in this study.

The limitations of this study are the following:

a. The small sample size obtained for an epidemiological study due to the lack of participation of the institution epidemiologists, which prevented identifying more ZIKV, DENV, or CHIKV cases related to GBS.

b. Controls were matched by age, gender and place of residence. Unfortunately, although over 75% of the population in Mexico is considered to be in a situation of poverty and vulnerability, socioeconomic level was not taken into account.

c. The selection of controls with no fever might have generated an overestimation of the exposure (ZIKV and DENV) in cases and an underestimation in controls. This fact should be considered for future studies, although it was recently shown that a study that only included GBS cases without febrile illness in the last 5 days before the onset of GBS was significantly different from controls without fever [27].

d. Although patients and interviewers did not take into account whether the recruited cases had presented prior infections related to any of the arboviruses, the interviews to identify prior diseases or symptomatology—of which individuals might be unaware—could be considered as another limitation of the study.

e. Other causes of GBS were not considered, such as *Campylobacter jejuni*, which causes acute diarrheal disease and can be related to more than 50% of GBS cases in Mexico. Respiratory viruses associated with the presence of GBS were also not considered.

In conclusion, the present study demonstrates laboratory evidence of ZIKV infection associated with GBS as well as laboratory evidence of the association of GBS with both ZIKV and DENV (flavivirus) infection. No association between GBS and CHIKV infection was found, nor were co-infections demonstrated among these three viruses. On the other hand, GBS cases associated with ZIKV showed a more severe clinical picture (more significant impairment of functional status and incapacity, and higher percentage of mechanical ventilation). Finally, the symptoms of ZIKV suspected disease were observed before the development of GBS, which is consistent with other studies published previously and supports the association between GBS and ZIKV infection.

## Supporting information

**S1 STROBE Statement Checklist.**
(DOCX)

**S1 Laboratory and Clinical database.**
(XLSX)

## Acknowledgments

The authors would like to thank Dr. James Sejvar, Dr. Rosalía Lira Carmona and Dr. Miguel Angel Villasís Keever for their important commentaries to the manuscript. We also would like to thank Anel Cantera Salinas, Osvaldo Jiménez Pacheco, Vidaris Toledo Arrazola and Mario Villegas Rivera, who helped us to produce the clinical database.

## Author Contributions

**Conceptualization:** Israel Grijalva, Concepción Grajales-Muñiz, César González-Bonilla, Victor Hugo Borja-Aburto, José Guerrero-Cantera.

**Data curation:** Martín Paredes-Cruz, José Guerrero-Cantera.

**Formal analysis:** Israel Grijalva, Martín Paredes-Cruz, José Guerrero-Cantera, Alfonso Vallejos-Parás, Porfirio Hernández-Bautista, Lumumba Arriaga-Nieto.

**Funding acquisition:** Israel Grijalva.

**Investigation:** Israel Grijalva, Concepción Grajales-Muñiz, César González-Bonilla, Victor Hugo Borja-Aburto, Martín Paredes-Cruz, José Guerrero-Cantera, Joaquín González-Ibarra, Alfonso Vallejos-Parás, Teresita Rojas-Mendoza, Clara Esperanza Santacruz-Tinoco, Porfirio Hernández-Bautista, Lumumba Arriaga-Nieto, Ma Guadalupe Garza-Sagástegui, Ignacio Vargas-Ramos, Ana Sepúlveda-Núñez, Omar Israel Campos-Villarreal, Roberto Corrales-Pérez, Mallela Azuara-Castillo, Elsa Sierra-González, José Alfonso Meza-Medina, Bernardo Martínez-Miguel, Gabriela López-Becerril, Jessica Ramos-Orozco, Tomás Muñoz-Guerrero, María Soledad Gutiérrez-Lozano, Arlette Areli Cervantes-Ocampo.

**Methodology:** Israel Grijalva, Victor Hugo Borja-Aburto, Martín Paredes-Cruz, José Guerrero-Cantera.

**Project administration:** Israel Grijalva, Gabriela López-Becerril.

**Supervision:** Israel Grijalva, Concepción Grajales-Muñiz, César González-Bonilla, Martín Paredes-Cruz, José Guerrero-Cantera, Joaquín González-Ibarra, Teresita Rojas-Mendoza, Clara Esperanza Santacruz-Tinoco, Porfirio Hernández-Bautista, Lumumba Arriaga-Nieto, Ma Guadalupe Garza-Sagástegui, Gabriela López-Becerril.

**Writing – original draft:** Israel Grijalva, Martín Paredes-Cruz, José Guerrero-Cantera.

**Writing – review & editing:** Israel Grijalva, Concepción Grajales-Muñiz, César González-Bonilla, Victor Hugo Borja-Aburto.

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
