## [Decision Letter · Decision Letter 0]

2 Mar 2020

Dear Dr. Grijalva,

Thank you very much for submitting your manuscript "Zika and dengue but not chikungunya are associated with Guillain-Barré syndrome in Mexico: a case-control study" for consideration at PLOS Neglected Tropical Diseases. As with all papers reviewed by the journal, your manuscript was reviewed by members of the editorial board and by several independent reviewers. In light of the reviews (below this email), we would like to invite the resubmission of a significantly-revised version that takes into account the reviewers' comments. 

We cannot make any decision about publication until we have seen the revised manuscript and your response to the reviewers' comments. Your revised manuscript is also likely to be sent to reviewers for further evaluation.

Sincerely,

Maya Williams

Associate Editor

Amy Morrison

Deputy Editor

Reviewer's Responses to Questions

**Key Review Criteria Required for Acceptance?**

**Methods**

-Are the objectives of the study clearly articulated with a clear testable hypothesis stated?

-Is the study design appropriate to address the stated objectives?

-Is the population clearly described and appropriate for the hypothesis being tested?

-Is the sample size sufficient to ensure adequate power to address the hypothesis being tested?

-Were correct statistical analysis used to support conclusions?

-Are there concerns about ethical or regulatory requirements being met?

Reviewer #1: Authors used methods used in several other countries in Latin America and elsewhere to study the association of flaviviruses and GBS. One limitation is the use of IgMs which may not be enough to differentiate Zika and Dengue. In the past, PRNTs have been obtained to fully differentiate immunologic evidence of recent flavivirus infection.

Reviewer #2: The methods section should be expanded and statistical analysis methods further clarified.

Definition of acute cases versus recent infections not clear. Clarify if there are overlaps. Detail detection limits of instruments used for testing (specificity and timing).

Reviewer #3: - Are the objectives of the study clearly articulated with a clear testable hypothesis stated? YES

-Is the study design appropriate to address the stated objectives? NO

-Is the population clearly described and appropriate for the hypothesis being tested? NO

-Is the sample size sufficient to ensure adequate power to address the hypothesis being tested? NO

-Were correct statistical analysis used to support conclusions? NO

-Are there concerns about ethical or regulatory requirements being met? YES

SEE SUMMARY AND GENERAL COMMENTS BELOW

**Results**

-Does the analysis presented match the analysis plan?

-Are the results clearly and completely presented?

-Are the figures (Tables, Images) of sufficient quality for clarity?

Reviewer #1: Increased mortality may be due to the bias towards sicker patients being included in the study.

Reviewer #2: - Table 2 should be clarified (acute vs recent infection cases)

- Results in the text not consistent with those presented in Table

- Modifications to Table 5 suggested; an additional Table suggested

Reviewer #3: -Does the analysis presented match the analysis plan? NO

-Are the results clearly and completely presented? NO. Results from matched case-control studies must be presented as discordant/concordant matched sets.

-Are the figures (Tables, Images) of sufficient quality for clarity? NO. See above response regarding data presented in Tables. Figure 1 is unreadable. According to Results, it seems to be an epidemiological curve of arbovirus infection and GBS cases. Unfortunately, the figure is completely irrelevant for the purpose of the study.

**Conclusions**

-Are the conclusions supported by the data presented?

-Are the limitations of analysis clearly described?

-Do the authors discuss how these data can be helpful to advance our understanding of the topic under study?

-Is public health relevance addressed?

Reviewer #1: I would tone down a bit the clinical differences between Zika and dengue in light of unavailability of PRNTs. 

Otherwise it reads well.

Reviewer #2: - Limitations need to be expanded. Main limitations not considered

Reviewer #3: -Are the conclusions supported by the data presented? NO

-Are the limitations of analysis clearly described? NO

-Do the authors discuss how these data can be helpful to advance our understanding of the topic under study? NO

-Is public health relevance addressed? SOMEHOW.

**Editorial and Data Presentation Modifications?**

Reviewer #1: Line 365: Verify spelling of dr. Styczynski last name.

Line 370: Please clarify meaning of ...the biological sample was not specific.

Line: 454: Verify meaning of sentence. researchers?

Reviewer #2: - The Tables could be better formatted' lack of consistency in digits after the decimal, which would make for a clearer read

- Some sentences could be improved for clarity (as marked on pdf)

- Copyediting would help the manuscript flow

Reviewer #3: - There are some other important limitations of this paper that were not included in the Reviewer Attachment.

The authors engaged in selective citation. They only cite articles that support their beliefs. In at least one case, the cited article does not support the authors claim (GeurtsvanKessel CH, et al.).

- As mentioned above, the epidemiological curve (Figure 1) is irrelevant.

- The authors mistakenly state that that an increase in GBS followed ZIKV outbreaks in Latin American countries. Previous studies have shown this was not the case. The claim that GBS increased in Mexico during 2016-2018 is unsupported by data. If that were the case, the increase could be easily explained by the implementation of GBS active surveillance, which started in 2015. At least one study (José Luis Soto-Hernández et al. doi: 10.3389/fneur.2019.00435) shows no increase in the incidence of GBS in 2016.

- Some studies cited in support of a Zika-GBS association actually do not show such association.

- Needed references are missing through the paper.

- In several statements, authors confound temporal concurrence (coincidence) with statistical association.

- Even if ZIKVI caused GBS, the management of a patient with GBS should be based on his/her clinical characteristics, not on the risk factors that may have led to the development of the disease.

- Researchers do not need to give consent to participate as investigators.

**Summary and General Comments**

Reviewer #1: Overall well written paper. Easy to read. Important contribution to he GBS/flavivirus literature.

Reviewer #2: This is a very important manuscript with public health significance and contributes to an understanding of the epidemiology of GBS associated with ZIKV, CHIKV and DENV in Mexico.

Reviewer #3: Reviewer comments

I have read with great interest Grijalva et al.’s manuscript. They recruited a sample of 97 cases of Guillain-Barré syndrome (GBS) and 184 age, gender, and residence-matched controls without GBS. They found that acute Zika virus infection (ZIKVI) and dengue virus infection (DVI) increased the risk of GBS by 16.45 times (95% CI: 2.03 – 133.56) and 4.15 times (95% CI: 0.74 − 23.1), respectively. When combining both exposures, the risk of GBS increased 6.35 times (95% CI: 1.99 – 20.28). They concluded that there is an evident association between these infections and the risk of GBS. On the other hand, they compared the severity of GBS in infected and non-infected cases and concluded that cases with ZIKVI, but not those with DVI, have a more severe clinical picture, and recommended that ZIKVI status should be taken into account when treating patients with GBS. 

Grijalva et al.’s is the second largest in a small group of case-controls studies of the association between ZIKVI and GBS risk. In spite of the obvious efforts of the investigators in conducting this study, its findings contribute little to clarify the possible causal nature of an association between arbovirus infections and risk of GBS. Indeed, like previous studies based on surveillance systems, this study has fundamental flaws that invalidate its findings. These flaws are described below.

Major Flaws

Detection bias occurs when the exposure of interest –ZIKVI or DVI− inﬂuences the diagnosis of the outcome −GBS. This type of bias is present in all studies of the association of GBS with ZIKVI/ and DVI that are based on GBS cases recruited from surveillance systems in Latin America. No country in Latin America had GBS surveillance before March 2016, when WHO/PAHO recommended the implementation of Zika-associated GBS surveillance.1 Unfortunately, without adequate evidence of a causal association, WHO/PAHO declared GBS was the most frequent neurological complication of ZIKVI and recommended looking for evidence of GBS in all ZIKVI cases and vice versa. These recommendations were adopted by the IMSS and the Mexican Ministry of Health.2-4 In fact, according to the IMMS Guidelines for the management of ZIKVI,4 this recommendation had been implemented in the IMMS –the source of GBS cases for Grijalva et al.’s study− at least since November, 2014. Some of the authors in Grijalva et al.’s paper co-authored the IMSS Guidelines. The recommendation about GBS surveillance was also incorporated in the national guidelines for surveillance for vaccine preventable diseases,2 under acute flaccid paralysis. Although surveillance guidelines do not directly call for the search of DVI in cases of GBS and vice versa, this is unavoidable due to the difficulties in differentiating DVI and ZIKVI, from both the clinical and laboratory perspective.

The search for ZIKVI and DVI in cases of GBS, and vice versa, recommended in surveillance guidelines, results in an overestimation of the frequency of the exposure –ZIKVI/DVI− in GBS cases selected from a surveillance system. In consequence, if ZIKVI and DVI were not associated to GBS incidence, they will appear to increase the risk if GBS cases were selected from the IMSS surveillance system. Correspondingly, if ZIKVI and DVI increased the risk of GBS, the increase in risk would be overestimated. 

This is not just a rational argument, as it is supported by data. Indeed, the magnitude of detection bias in Grijalva et al.’s study can be estimated using the incidence of Zika virus infection in Mexico (82 per 100,000)3 and the observed prevalence of ZIKVI in their sample of cases (8 out of 97).5 The smallest OR for ZIKVI-GBS that would have resulted in 8 out of 97 cases infected with Zika virus was 100.6, and the observed prevalence of ZIKVI among GBS cases was 11.4 time higher than expected.

Briefly, the causal effect of ZIKVI/DVI on the risk of GBS could not be identified –measured or estimated− in Grijalva et al.’s study, due to the presence of detection bias. The large observed ORs in this study could be easily explained by detection bias. Unfortunately, there is no way to overcome this limitation.

Selection bias: A fundamental principle in the design of case control studies is that case and control selection must be independent of the exposure. This means that in Grijalva et al.’s study, GBS cases and controls must have been selected independently of ZIKVI and DVI. As explained above, this principle was violated for the case group, which included cases of concurrent −coincidental− GBS and ZIKVI, instead of cases of just GBS. In addition, the principle was violated during the selection of the control group, because only patients without fever were eligible as controls. Excluding patients with fever from the control group results in an underestimation of the frequency of the exposure (ZIKVI/DVI) in this group, because fever is common in patients with these diseases. It is not surprising that Grijalva et al. found a strong effect of these infections in the risk of GBS, because their approach resulted in overestimation of the exposure (ZIKVI and DVI) in cases and underestimation in controls. If no association existed, Grijalva et al.’s study was bound to show a positive association. If a positive association existed, Grijalva et al.’s study was bound to overestimate that association. 

Another limitation of this study is the lack of details on eligibility and selection of controls. The only information provided was that controls were matched on state of residence, gender, and age to the cases, in a 2:1 ratio. Without further information, it is impossible to judge whether controls were a representative sample of the population from which the cases came from. For instance, were controls residing in high altitude places and those with hemostatic disorders eligible? Were patients with diseases potentially associated with the exposure excluded from the control group? Which diseases were excluded? How and where were eligible controls identified? In those cases where more than two matched controls were eligible for a given case, how were the two controls selected? Authors state they selected a non-probabilistic sample of controls. Unfortunately, unless the approach for control selection is explained it would be impossible to judge if it resulted in selection bias. 

Out of 1030 GBS cases, only 97 were included in this study. The only eligibility criterion for was having a 1-3 Brighton level of certainty. This corresponds to 90% of all cases being Brighton level 4, as compared to 1.1% in previous GBS series.5 This remarkable difference suggests the accuracy of the diagnosis of GBS at the IMSS was incredibly poor or that other undisclosed GBS eligibility criteria were applied in this study. Without knowing the eligibility criteria, no judgment on the validity of the findings could be made. 

In summary, the causal effect of ZIKVI/DVI on the risk of GBS could not be identified –measured or estimated− in Grijalva et al.’s study, due to the presence of uncorrectable selection bias. There is no way to overcome this limitation. 

Information bias:

Symptoms prior to the development of GBS were collected by interview. If cases were aware of they had GBS, they may have been trying, consciously or unconsciously, to concoct a reason why they got such a rare, serious disease. In consequence, they may have been more likely to report more symptoms and more severe symptoms than controls. In a similar fashion, if interviewers were aware of case/control status of the patient and the hypothesis being tested, they may have probed more deeply for ZIKVI and DVI symptoms in cases than in controls. Authors should describe how data were collected, to enable readers to judge the quality of their study. 

The fact that tests for ZIKVI and DVI were conducted after the diagnosis of GBS –up to 14 days−, makes reverse causality bias possible. In other words, it is possible that GBS cases got ZIKVI or DVI after they developed GBS. In those cases, ZIKVI and DVI would be irrelevant exposures, as they could not cause the GBS.

None of these likely sources of information bias could be remedy in this study. 

Confounding bias: Variables that are risk factors for GBS and are associated to ZIKVI and DVI could partly or completely explain the observed association between these diseases. For instance, patients from low socioeconomic status are more exposed to mosquito bites. This increases their risk of developing ZIKVI and DVI. They are also more likely to experience diarrheal diseases and acute respiratory infections, which are known to increase the risk of GBS. In fact, in a series of 17 prospective GBS cases from Mexico,6 75% were positive for Campylobacter jejuni. However, Grijalva et al.’s study does not account for these or other potential confounders. Indeed, based on data in Table 5, the odds ratios for acute diarrheal disease and acute respiratory infections were 14.00 and 3.40, respectively. Therefore, these two risk factors could partly or completely explain the observed association between ZIKVI/DVI and risk of GBS. 

Adjusting for confounding variables in this study is not possible, due to the small number of observations in strata defined by exposure, outcome, and confounder (see below). 

Model specification bias

A fundamental limitation of Grijalva et al.’s study is that the data analysis does not correspond to the study design. In other words, the analysis does not reflect the mechanism by which the data were generated. Matching in a case-control study introduces selection bias. A matched analysis must be conducted to correct for the selection bias resulting from matching. Although the authors provide insufficient information about their analytical approach, besides mentioning a couple of statistical tests, it is obvious they calculated ORs using simple 2×2 tables, as if the study were unmatched. 

It is not surprising the investigators conducted an unmatched analysis. Matched analyses are based on the concordance of matched sets. Concordant triples or pairs (in this case) are those matched sets where the case has the same value of the exposure as the controls. If the case and controls in a matched set are all exposed or all non-exposed, the matched set does not contribute to the estimation of the OR. Only discordant pairs or triples –those with at least one control with a different exposure value as the case− contribute to the estimation of the OR. By looking at Grijalva et al.’s Table 1, it is easy to see that only matched sets from Nuevo Leon and Veracruz included discordant pairs. Moreover, only one discordant matched set –from Veracruz−has a control with ZIKVI. In consequence, a matched analysis of this case-control study –the correct analysis, considering the design− would be impossible. A correct approach to analyze the data from this study would be using conditional logistic regression. The Stata code in the attached document reproduces the data in Table 1, for GBS and ZIKVI, and then runs a conditional logistic regression. As expected, Stata cannot find an estimate for the conditional OR, because there are 

too few data points.

Calculating ORs from 2×2 tables –the approach used by the investigators− not only leads to selection bias –because it ignores the matching−, but also fails due to the rather small number of data points. For instance, the OR for ZIKVI, from Table 1, would be (8/89)÷(1/183) = 16.45 (95% CI: 2.13, 733.64). The extremely large OR, with the excessively wide conﬁdence intervals indicates sparse data bias, instead of a strong effect of ZIKVI on GBS risk.7 Basically, the fact that there is only one control that was exposed makes the OR extremely large and unreliable. A way to put this in simple terms is that the amount of data they had was insufficient to figure out with any certainty what was going on. In the presence of sparse data, the OR grossly overestimate the true effect of the exposure on the outcome. Combining ZIKVI and DVI does not solve this problem. The average of two biased ORs (bottom of Table 2) is another biased OR. Moreover, combining two very different ORs (16.45 for ZIKVI and 4.15 for DVI) makes no mathematical, nor biological sense.

The problem of sparse data bias is even more marked in the comparisons of clinical characteristics of GBS patients with and without ZIKVI (Tables 4 and 5). For instance, the claim that mortality in GBS cases with ZIKVI infection is higher than GBS cases without ZIKVI was based on only one death in cases with ZIKVI.

Using Stata, I was unable to replicate the confidence intervals for ORs reported by the authors.

Briefly, the approach to the data analysis used in this study is incorrect and results in an overestimate of the effect of ZIKVI and DVI on GBS risk. A correct analysis, based on a conditional likelihood, is impossible. Both problems are due to the scarcity of the data.

References

1. Pan_American_Health_Organization. Guidelines for zika virus disease and complications surveillance. 2016

2. Martínez JCR, Guzmán NIL, Tello NMR, López FJA, García MAG, Almeida MdCG, Álvarez GL, Patoni MEV, Martínez IL, Longoria BT, Flores BO, Pedroza JFR, Ramírez EC, Sánchez EP, López LÁS, Bautista PH, Bocanegra VG, Reyes MÁA, Quiroz ZG, Carrillo JMH, Vargas YM, Salinas JS, Ochoa AR, Arias EH, Ibánez CRC, Gallego MdCI, Silva LdlC, Domínguez CVAS, Colima NM, Gómez. LFO. Manual de procedimientos estandarizados para la vigilancia epidemiológica de las enfermedades prevenibles por vacunación. 2018

3. Grajales-Muñiz C, Borja-Aburto VH, Cabrera-Gaytán DA, Rojas-Mendoza T, Arriaga-Nieto L, Vallejos-Parás A. Zika virus: Epidemiological surveillance of the mexican institute of social security. PLoS One. 2019;14:e0212114

4. Barrera-Cruz A, Díaz-Ramos RD, López-Morales AB, Grajales-Muñiz C, Viniegra-Osorio A, Zaldívar-Cervera JA, Arriaga-Dávila JdJ. Lineamientos técnicos para la prevención, diagnóstico y tratamiento de la infección por virus zika. Rev. Med. Inst. Mex. Seguro Soc. 2016;54:13

5. Bautista LE. Zika virus infection and risk of guillain-barré syndrome: A meta-analysis. J Neurol Sci. 2019;403:99-105

6. Soto-Hernández JL, Ponce de León Rosales S, Vargas Cañas ES, Cárdenas G, Carrillo Loza K, Díaz-Quiñonez JA, López-Martínez I, Jiménez-Corona M-E, Ruiz-Matus C, Kuri Morales P. Guillain–barré syndrome associated with zika virus infection: A prospective case series from mexico. Front. Neurol. 2019;10

7. Greenland S, Schwartzbaum JA, Finkle WD. Problems due to small samples and sparse data in conditional logistic regression analysis. Am J Epidemiol. 2000;151:531-539

PLOS authors have the option to publish the peer review history of their article (what does this mean?). If published, this will include your full peer review and any attached files.

Reviewer #1: No

Reviewer #2: Yes: Ariadna Capasso

Reviewer #3: Yes: Leonelo E Bautista
---

## [Decision Letter · Decision Letter 1]

8 Jul 2020

Dear Dr. Grijalva,

Thank you very much for submitting your manuscript "Zika and dengue but not chikungunya are associated with Guillain-Barré syndrome in Mexico: a case-control study" for consideration at PLOS Neglected Tropical Diseases. As with all papers reviewed by the journal, your manuscript was reviewed by members of the editorial board and by several independent reviewers. The reviewers appreciated the attention to an important topic. Based on the reviews, we are likely to accept this manuscript for publication, providing that you modify the manuscript according to the review recommendations. 

Sincerely,

Maya Williams

Associate Editor

Amy Morrison

Deputy Editor

Reviewer's Responses to Questions

**Key Review Criteria Required for Acceptance?**

**Methods**

-Are the objectives of the study clearly articulated with a clear testable hypothesis stated?

-Is the study design appropriate to address the stated objectives?

-Is the population clearly described and appropriate for the hypothesis being tested?

-Is the sample size sufficient to ensure adequate power to address the hypothesis being tested?

-Were correct statistical analysis used to support conclusions?

-Are there concerns about ethical or regulatory requirements being met?

Reviewer #1: I believe the authors have addressed all the points I brought up in the first round of reviews.

Reviewer #2: Yes

Reviewer #3: All my comments are in the attached document. My original qualifications have not changed

**Results**

-Does the analysis presented match the analysis plan?

-Are the results clearly and completely presented?

-Are the figures (Tables, Images) of sufficient quality for clarity?

Reviewer #1: I believe the authors have addressed all the points I brought up in the first round of reviews.

Reviewer #2: Yes. The paper is much improved from the previous version.

Reviewer #3: All my comments are in the attached document. My original qualifications have not changed

**Conclusions**

-Are the conclusions supported by the data presented?

-Are the limitations of analysis clearly described?

-Do the authors discuss how these data can be helpful to advance our understanding of the topic under study?

-Is public health relevance addressed?

Reviewer #1: I believe the authors have addressed all the points I brought up in the first round of reviews.

Reviewer #2: Need to modify the conclusions to get rid of causal language. Causality cannot be established with this study design.

Reviewer #3: All my comments are in the attached document. My original qualifications have not changed

**Editorial and Data Presentation Modifications?**

Reviewer #1: I believe the authors have addressed all the points I brought up in the first round of reviews.

Reviewer #2: (No Response)

Reviewer #3: All my comments are in the attached document. My original qualifications have not changed

**Summary and General Comments**

Reviewer #1: I believe the authors have addressed all the points I brought up in the first round of reviews. They have also been very gracious in their acceptance of edits and suggestions from co-reviewers. Overall, this is a contribution to the literature from bother geographical area increasing the plausibility of the association of flaviviruses and GBS.

Reviewer #2: The manuscript is much improved and much clearer than the first version. It contributes with findings from an understudied disease and population. It requires one more round of editing to make sure there are no errors (e.g. check statistical significance, direction of < and >, etc), remove causal language in discussion and rather focus on association, and rectify some grammatical erroes (some of suggested modifications were marked on the pdf).

Reviewer #3: This study have major design, data, and analytical flaws that invalidate its findings. These flaws cannot be addressed through data analysis, because the exposure and outcome were forced to be correlated during the diagnosis of the outcome and the evaluation of the exposure. There is no adjustment for confounding factors and selection bias was introduced by excluding controls with fever. Other major flaw is that the study is matched and this introduces a selection bias that can only be addressed by using a matched analysis. Unfortunately, a matched analysis is impossible due to very small number of discordant set. The later problem, the lack of data, result in sparse data bias, i.e. extremely large and unreliable odds ratios that the authors wrongly misinterpret as associations.

PLOS authors have the option to publish the peer review history of their article (what does this mean?). If published, this will include your full peer review and any attached files.

Reviewer #1: No

Reviewer #2: Yes: Ariadna Capasso

Reviewer #3: Yes: Leonelo E Bautista
---

## [Decision Letter · Decision Letter 2]

5 Oct 2020

Dear Dr. Grijalva,

We are pleased to inform you that your manuscript 'Zika and dengue but not chikungunya are associated with Guillain-Barré syndrome in Mexico: a case-control study' has been provisionally accepted for publication in PLOS Neglected Tropical Diseases.

Best regards,

Maya Williams

Associate Editor

Amy Morrison

Deputy Editor

Reviewer's Responses to Questions

**Key Review Criteria Required for Acceptance?**

**Methods**

-Are the objectives of the study clearly articulated with a clear testable hypothesis stated?

-Is the study design appropriate to address the stated objectives?

-Is the population clearly described and appropriate for the hypothesis being tested?

-Is the sample size sufficient to ensure adequate power to address the hypothesis being tested?

-Were correct statistical analysis used to support conclusions?

-Are there concerns about ethical or regulatory requirements being met?

Reviewer #2: The objectives and design are clear. There are no ethical concerns.

**Results**

-Does the analysis presented match the analysis plan?

-Are the results clearly and completely presented?

-Are the figures (Tables, Images) of sufficient quality for clarity?

Reviewer #2: The results comply with PLOS criteria.

**Conclusions**

-Are the conclusions supported by the data presented?

-Are the limitations of analysis clearly described?

-Do the authors discuss how these data can be helpful to advance our understanding of the topic under study?

-Is public health relevance addressed?

Reviewer #2: The conclusions are relevant to the data. The paper is of public health relevance.

**Editorial and Data Presentation Modifications?**

Reviewer #2: Accept.

**Summary and General Comments**

Reviewer #2: The authors have sufficiently addressed previous reviewer issues.

PLOS authors have the option to publish the peer review history of their article (what does this mean?). If published, this will include your full peer review and any attached files.

Reviewer #2: **Yes: **Ariadna Capasso

---

## [Editor Report · Acceptance letter]

26 Nov 2020

Dear Dr. Grijalva,

We are delighted to inform you that your manuscript, "Zika and dengue but not chikungunya are associated with Guillain-Barré syndrome in Mexico: a case-control study," has been formally accepted for publication in PLOS Neglected Tropical Diseases.

Best regards,

Shaden Kamhawi

co-Editor-in-Chief

Paul Brindley

co-Editor-in-Chief
